# Assessing hearing loss in older adults with a single question and person characteristics; Comparison with pure tone audiometry in the Rotterdam Study

**Berthe C. Oosterloo**[1,2]*, **Nienke C. Homans**[1], **Rob J. Baatenburg de Jong**[1], **M. Arfan Ikram**[2], **A. Paul Nagtegaal**[1], **André Goedegebure**[1,2]

1 Department of Otorhinolaryngology and Head and Neck Surgery, Erasmus MC, The Netherlands,
2 Department of Epidemiology, Erasmus MC, The Netherlands

* b.oosterloo@erasmusmc.nl

## Abstract

### Introduction

Hearing loss (HL) is a frequent problem among the elderly and has been studied in many cohort studies. However, pure tone audiometry—the gold standard—is rather time-consuming and costly for large population-based studies. We have investigated if self-reported hearing loss, using a multiple choice question, can be used to assess HL in absence of pure tone audiometry.

### Methods

This study was performed within 4,906 participants of the Rotterdam Study. The question (in Dutch) that was investigated was: 'Do you have any difficulty with your hearing (without hearing aids)?'. The answer options were: 'never', 'sometimes', 'often' and 'daily'. Mild hearing loss or worse was defined as $PTA_{0.5-4}$ (Pure Tone Average 0.5, 1, 2 & 4 kHz) $\geq$20dBHL and moderate HL or worse as $\geq$35dBHL. A univariable linear regression model was fitted with the $PTA_{0.5-4}$ and the answer to the question. Subsequently, sex, age and education were added in a multivariable linear regression model. The ability of the question to classify HL, accounting for sex, age and education, was explored through logistic regression models creating prediction estimates, which were plotted in ROC curves.

### Results

The variance explained ($R^2$) by the univariable regression was 0.37, which increased substantially after adding age ($R^2$ = 0.60). The addition of sex and educational level, however, did not alter the $R^2$ (0.61). The ability of the question to classify hearing loss, reflected in the area under the curve (AUC), was 0.70 (95% CI 0.68, 0.71) for mild hearing loss or worse and 0.86 (95% CI 0.85, 0.87) for moderate hearing loss or worse. The AUC increased substantially when sex, education and age were taken into account (AUC mild HL: 0.73 (95%CI 0.71, 0.75); moderate HL 0.90 (95%CI 0.89, 0.91)).

**Data Availability Statement:** Because of restrictions based on privacy regulations and

informed consent of the participants, data cannot be made freely available in a public repository. Data can be obtained on request. Requests should be directed toward the management team of the Rotterdam Study (secretariat.epi@erasmusmc.nl), which has a protocol for approving data requests for researchers who meet the criteria for access to confidential data.

**Funding:** Cochlear Ltd. provided partial funding for this study to AG. Heinsius Houbolt fonds also provided support for authors BCO and AG for this study. The specific role of these authors are articulated in the 'author contributions' section. The funders had no role in study design, data collection and analysis, decision to publish, or preparation of the manuscript.

**Competing interests:** The authors have read the journal's policy and the authors of this manuscript have the following competing interests: AG received funding from Cochlear Ltd. BCO and AG also received funding from Heinsius Houbolt fonds. This does not alter our adherence to PLOS ONE policies on sharing data and materials. There are no patents, products in development or marketed products to declare.

## Conclusion

Self-reported hearing loss using a single question has a good ability to detect hearing loss in older adults, especially when age is accounted for. A single question cannot substitute audiometry, but it can assess hearing loss on a population level with reasonable accuracy.

## Introduction

Age-related hearing loss (ARHL) is considered to be one of the most common chronic disorders among the elderly [1–3]. Generally, world-wide life expectancy increases, resulting in an upsurge of age-related health problems, including hearing loss. The prevalence of hearing loss in people over 65 years old ranges from 29% to 47%, based on the WHO definition of the pure tone average over 0.5, 1, 2 and 4 kHz in the better ear ($PTA_{0.5-4}$) >25 decibel hearing level (dBHL) [1–4]. Its impact is substantial, as it is associated with social withdrawal, cognitive decline and depression [5–8]. The gold standard for measuring hearing loss is pure tone audiometry, which is not always available in large population-based studies. Self-reporting scales regarding hearing are more readily available and are therefore often used in the absence of pure tone audiometry [9, 10].

The assessment of subjective hearing loss in an ageing population through a single question or through questionnaires has been extensively investigated over the last two decades. Self-reporting is suitable for estimating the prevalence of hearing impairment, particularly for moderate to severe hearing loss [11]. The most commonly used, validated questionnaire is the hearing handicap inventory for the elderly (HHIE) [10, 12], and its screening version (HHIE-s) [13]. In addition, there are several studies that investigated a single question format. A review of 10 studies published between 1990 and 2004, concluded that a single question is able to identify hearing loss (especially moderate and severe) in adults over 60 years of age [14]. This review concludes that a single question is an acceptable substitute when audiometric measurements are not available, for example in epidemiological studies [14]. An overall linear relationship between self-reported hearing loss and pure tone audiometry is reported, although trends towards underestimation of hearing loss can be noted, especially in subjects younger than 60 years [15].

However, several issues remain to be addressed when using a question to estimate hearing loss in a general elderly population. First of all, many studies also included younger adults (from 20 years old), which has a significant influence on the outcome. At this young age, hearing is generally good and will be adequately reported as no hearing problems, in contrast to an elderly population characterized by a much higher prevalence of hearing loss. Secondly, most conclusions are based on the identification of ($PTA_{0.5-4}$) $\geq$ 40 dBHL, thereby excluding slighter hearing loss. Slight hearing loss may already lead to communication problems in everyday situations, but is harder to identify as such on an individual level. It would therefore be of added value if a question could also differentiate between various degrees of hearing loss. A third point that deserves attention is how individual characteristics, including age and sex, may influence the outcome. Most studies report whether certain subgroups of individuals are better or worse at judging their hearing capacity. However, none of the studies have investigated how these easily obtained participant characteristics may actually be used to improve the estimation of hearing loss using a single question format.

The aim of our study was to investigate the reliability of a single question in assessing hearing loss in a large, ageing population. We used two thresholds for $PTA_{0.5-4}$: >20 dBHL (mild

or worse) and >35 dBHL (moderate or worse). Additionally, we investigated the ability of the question to assess hearing loss when adding individual characteristics, including age, sex and level of education.

## Methods

### Study subjects

This study was part of the Rotterdam Study, a prospective cohort study that has been ongoing since 1990, in which risk factors for common diseases are investigated in an ageing population [16, 17]. We report on data collected between February 2011 and June 2016. People aged 45 years and older were invited for participation via the population registry of Ommoord, a suburb of Rotterdam, The Netherlands. A total of 4,906 participants underwent both a home interview and pure tone audiometry by an experienced audiometrist in a dedicated research center in Ommoord. When individuals were screened at multiple time points, only the most recent one was taken into analysis.

The Rotterdam Study has been approved by the medical ethics committee of the Erasmus MC (registration number MEC 02.1015) and by the Dutch Ministry of Health, Welfare and Sport (Population Screening Act WBO, license number 1071272-159521-PG). The Rotterdam Study has been entered into the Netherlands National Trial Register (www.trialregister.nl) and into the World Health Organization International Clinical Trials Registry Platform (who.int/ictrp/network/primary/en/) under shared catalog number NTR6831. All participants provided written informed consent to participate in the study.

### Interview

Each participant underwent an extensive home interview on health, background and environmental factors, before visiting the research center [16]. Participants were asked the following question (in Dutch), similar to that used in previous studies [18]: 'Do you have any difficulty with your hearing (without hearing aids)?' with answers ranging on a 4-point scale: 'No, I always hear everything', 'Yes, sometimes I do not hear what is being said', 'Yes, I regularly do not hear what is being said' or 'Yes, I almost never hear what is being said'. Amongst many other parameters, highest achieved educational level was noted, using the UNESCO classification [19].

### Pure tone audiometry

For all audiometric measurements, a clinical audiometer was used (Decos audiology workstation, version 210.2.6, with AudioNigma interface; Decos Audiology, Inc., Peachtree City, GA) with TDH-39P earphones. Measurements were performed in a professional soundproof booth. Hearing thresholds were set at the intensity level at which the tone was heard in 2 out of 3 ascents, according to the ISO standard 8253–1 [20]. If no response was obtained, even at maximum stimulation level for that given frequency, the threshold was set at 5 dB above maximum stimulation level. Air-conduction thresholds were obtained at octave frequencies 0.25, 0.5, 1, 2, 4 and 8 kHz. The subjectively better ear was measured first. For the analyses, we used a mid-frequency average at 0.5, 1, 2 and 4 kHz ($PTA_{0.5-4}$) in the better ear and cut-offs proposed by the WHO [4, 21, 22].

### Statistical analysis

Data management and analyses were done using IBM SPSS Statistics 24. A univariable linear regression model was created to investigate the association between the interview answers and

$PTA_{0.5-4}$ with forward selection of the independent variables sex, level of education and age. These three independent variables were chosen because they are easily assessed in a clinical situation. Independent variables were tested for their contribution to the regression with a Likelihood Ratio Test. The independent variable with the largest change in goodness of fit ($R^2$) was first taken into the model and the other independent variables were added subsequently. The beta's standardized regression coefficients were reported to show the effect sizes of the associations found.

For our next analyses, interview answers were dichotomized. For mild hearing loss (or worse), defined as $PTA_{0.5-4} \geq 20$ dBHL in the better ear, only the first answer option was considered negative ('No, I always hear everything') and all other answers were considered positive. For moderate to severe hearing loss (or worse), defined as $PTA_{0.5-4} \geq 35$ dBHL in the better ear, the first two answer options ('No, I always hear everything' and 'Yes, sometimes I do not hear what is being said') were considered negative and the other answers were considered positive [4]. With these dichotomized outcomes, we calculated sensitivity, specificity as well as negative and positive predictive values. Subsequently, a logistic regression model was fitted to calculate prediction estimates. Hearing loss was the outcome variable and the dichotomized interview answers were the investigated predictive values. The same independent variables from the final linear regression model were also used in the logistic regression model. The prediction estimates were used to plot receiver operating characteristic (ROC) curves. The area under the curve (AUC) was calculated to determine the discriminatory value of the question for hearing loss.

**Secondary analysis.** The full logistic regression model was repeated to calculate prediction estimates in a dataset stratified on age (<65 years of age, 65–80 years of age and >80 years of age). These prediction estimates were used to find the discriminatory value of self-reported hearing loss within each age category.

## Results

### Hearing loss as a continuous variable

Characteristics of the study population (n = 4,906) are listed in Table 1. We found a higher average hearing threshold in men, older participants and participants with only primary school level education (S1 Table). Distribution of the answers to the question was categorized per 20 dBHL hearing loss (Fig 1). As the hearing loss increased, the answers shifted from 'no' or 'sometimes a problem' to 'often' or 'almost always a problem'. Above 40 dBHL ($PTA_{0.5-4}$) hearing loss, almost all participants confirmed to have some degree of hearing problems.

Univariable linear regression analysis showed an increase of the $PTA_{0.5-4}$ hearing threshold by 10.5 dB (95%CI 10.07, 10.85) for each step up in subjective difficulty in hearing (Table 2). Adding either sex, education or age led to a significant improvement of the regression (p<0.0001). The largest increase of the explained variance was seen for the factor "age" ($R^2$ increase from 0.37 to 0.60, Fig 2). The subsequent addition of education and sex to the regression improved the model (p<0.0001), although the explained variance remained almost unchanged ($R^2$ 0.61).

### Hearing loss as a dichotomous variable

Table 3 shows the 2x2 tables for the dichotomized answers and the presence of hearing loss ($PTA_{0.5-4}$) (mild: $\geq 20$ dBHL / moderate: $\geq 35$dBHL). Prevalence of subjective mild hearing loss was 51.3% against 52.6% for objective mild hearing loss, while prevalence of subjective moderate hearing loss was 17.8% against 19.8% for objective moderate hearing loss.

**Table 1. Characteristics of participants who had been asked the question: 'Do you have any difficulty with your hearing?'.** Hearing thresholds were averaged over 0.5, 1, 2 & 4 kHz ($PTA_{0.5-4}$).

| N | 4,906 |
|---|---|
| Mean age, years (SD) | 69.6 (9.8) |
| Age range, years | 51.4–100.7 |
| Female, % | 56.3 |
| Average hearing threshold, dBHL (SD) | 24.5 (13.9) |
| Mild hearing loss or worse ($PTA_{0.5-4}$), % | |
| No (<20 dBHL) | 47.4 |
| Yes ($\geq$ 20dBHL) | 52.6 |
| Moderate hearing loss or worse ($PTA_{0.5-4}$), % | |
| No (< 35 dBHL) | 80.2 |
| Yes ($\geq$ 35 dBHL) | 19.8 |
| Level of education, % | |
| Primary | 7.8 |
| Lower | 38.9 |
| Intermediate | 30.0 |
| Higher | 23.3 |
| Answer to the question, % | |
| Never | 48.7 |
| Sometimes | 33.6 |
| Regularly | 15.6 |
| Often | 1.9 |

The numbers in the 2x2 table were used to calculate sensitivity (mild hearing loss 69.9%, moderate hearing loss 54.8%), specificity (mild hearing loss 69.2%, moderate hearing loss 91.4%), positive predictive value (mild hearing loss 71.5%, moderate hearing loss 61.1%) and negative predictive value (mild hearing loss 67.4%, moderate hearing loss 89.1%).

A logistic regression model was used to identify the presence or absence of hearing loss at $\geq$ 20 dBHL and $\geq$ 35dBHL ($PTA_{0.5-4}$), based on the question. Sex, education and age were taken into account as independent variables. The ability to identify mild hearing loss ($PTA_{0.5-4} \geq$ 20 dBHL), reflected in the AUC, was 0.70 (95% CI 0.68, 0.71). This increased when sex, education and age were taken into account (AUC: 0.86 (95% CI 0.85, 0.87)). The AUC for the identification of moderate hearing loss was 0.73 (95%CI 0.71, 0.75), which increased to 0.90 (95%CI 0.89, 0.91) with the addition of sex, education and age.

In a secondary analysis, a logistic regression model, adjusted for age, sex and education, was repeated in a dataset stratified in 3 age categories (<65 years of age, 65–80 years of age and >80 years of age). The AUC for mild hearing loss ($PTA_{0.5-4} \geq$ 20 dBHL) was highest for the oldest and lowest for the youngest age group (AUC <65 years 0.75 (95%CI 0.73, 0.78); 65–80 years 0.79 (95%CI 0.77,0.81); >80 years 0.81 (95%CI 0.76, 0.85)). For moderate hearing loss, the AUC was highest for the youngest age group and decreased with increasing age (AUC <65 years 0.86 (95%CI 0.80, 0.92); 65–80 0.83 (95%CI 0.81, 0.85); >80 years 0.79 (95%CI 0.76, 0.82)).

## Discussion

In this study, we investigated the ability of the question 'Do you have any difficulty with your hearing (without hearing aids)?' to classify the severity of hearing loss measured by pure tone audiometry in an ageing population, using a four-category response. We have shown that this

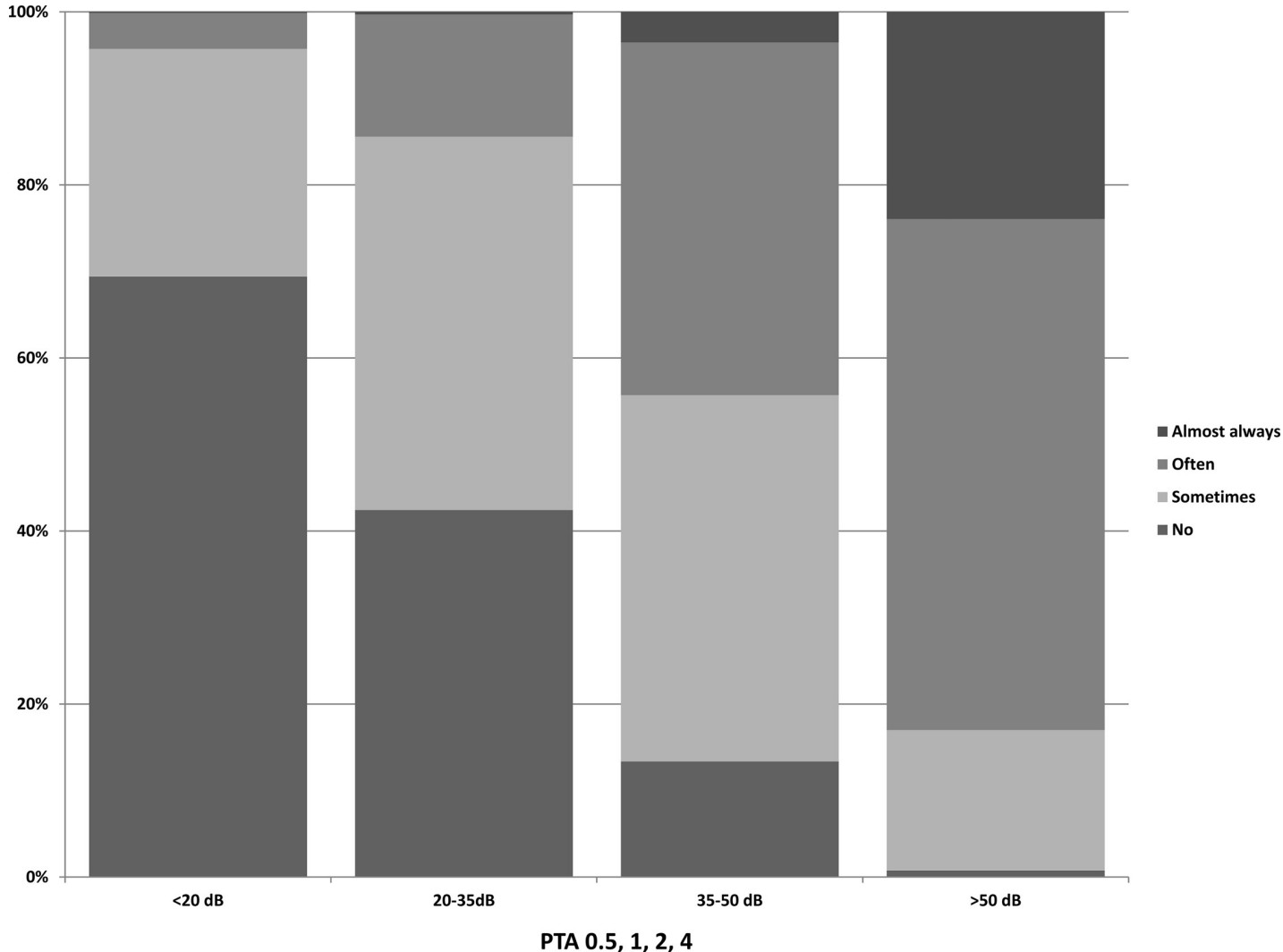

**Fig 1. Distribution of answers to the question: 'Do you have any difficulty with your hearing?'.** Per 15dB hearing loss (PTA$_{0.5-4}$) in the better ear.

single question can be used to identify both mild and moderate hearing loss with reasonable accuracy. The ability of the question to identify hearing loss increases substantially when other factors are taken into account, with age being the most important one.

Our results are in line with the increasing support in literature for using a single question as an estimator for hearing loss in absence of pure tone audiometry [14, 15, 23–28]. There is a growing general interest in applying this concept to large population-based studies, for which time or other resources to perform audiometry are not available [29, 30]. We have shown that for these large population-based studies, a single question (adjusted for age, sex and highest achieved education) is a good surrogate for the actual hearing ability. Nevertheless, a single question might also be of value from a clinical perspective to identify populations at risk for hearing loss and should therefore be taken in consideration for screening purposes. Of course, one question testing is not meant to replace pure tone audiometry in the assessment of hearing in an individual.

In addition, we compared the ability of the same question to assess both mild (PTA$_{0.5-4}$ ≥20 dBHL) and moderate hearing loss (PTA$_{0.5-4}$ ≥35 dBHL). The predictive ability of the

**Table 2. Results from the linear regression analysis for the question: 'Do you have any difficulty with your hearing?'.** First, univariable analysis was done. Then each of the independent covariates were added, initially separately and later together. Beta's reflect the number of decibel change in the $PTA_{0.5-4}$ with each step up in degree of subjective hearing loss (never, sometimes, regularly, or often). $R^2$ is given as a measure of the goodness of fit of the model.

| | Intercept (95%CI) | Beta (95%CI) | $R^2$ |
|---|---|---|---|
| Univariable | 17.05 (16.64, 17.47) | 10.46 (10.07, 10.85) | 0.37 |
| + sex | 17.65 (17.10, 18.21) | 10.41 (10.02, 10.80) | 0.38 |
| + education | 19.74 (19.03, 20.45) | 10.37 (9.98, 10.75) | 0.39 |
| + age (/ year) | -30.66 (-32.45, -28.87) | 8.26 (7.94, 8.58) | 0.60 |
| +age + education | -29.02 (-30.96, -27.09) | 8.24 (7.92, 8.57) | 0.61 |
| + age + sex + education | -27.56 (-29.54, -25.59) | 8.16 (7.84, 8.49) | 0.61 |

question for hearing loss (when taking age, sex and educational level into account) was 88% for mild hearing loss and 92% for moderate hearing loss. This is slightly higher than the previously reported ability of the HHIE-s to detect hearing loss (cut-off point at 8), where the AUC was 79% and 86% for mild and moderate hearing loss, respectively [31]. A single question is thought to be at least as good as or better than the HHIE-s, in detecting both mild and moderate hearing loss [12, 31]. One of the reasons is that HHIE-s has a broader scope than identifying hearing loss, as the HHIE(-s) also measures the possible impact of hearing loss on daily life [9]. Generally, a single question identifies moderate hearing loss better than mild hearing loss [14, 31, 32]. Nevertheless, we found that, when some participant characteristics are taken into account, the single question is able to also identify mild hearing loss. This might be explained by the fact that we used a 4-category response, instead of a simple yes/no, which allows for distinguishing between different grades of auditory-performance problems. Therefore, it is advisable to use more than two categories when it is also important to identify mild hearing loss.

Self-reporting always comes with the risk of misclassification bias, resulting in under- and overestimation. Age, sex and educational level, amongst other factors, are shown to be associated with identification of self-reported hearing loss [24, 25, 33] and objective hearing loss and are easily assessable in any situation [1, 14, 15, 24, 33, 34]. We found age, sex and educational level to increase the ability of the question to identify hearing loss, with age being the most important factor by far. When accounting only for age, the answer to our single question explained 61% of the hearing threshold, an increase of 24%-points. Older participants appear to be better at reporting their limitations in hearing ability than younger participants. An explanation might be that it is socially more acceptable for older people to experience hearing impairment. Compensation mechanisms for hearing loss do not function as well in older as they do in younger persons [35]. The role of age in the correct identification of hearing loss may also be attributable to the difference in prevalence of hearing loss, as reflected in the stratified analyses. As hearing is generally good in younger people, moderate hearing loss is uncommon, underestimation of hearing impairment and its symptoms is hardly possible. There might even be a bigger chance of overestimation of the hearing loss because the younger people often find themselves in more challenging listening environments. In older people with more prevalent hearing loss, both under- and overestimation are possible. The oldest group

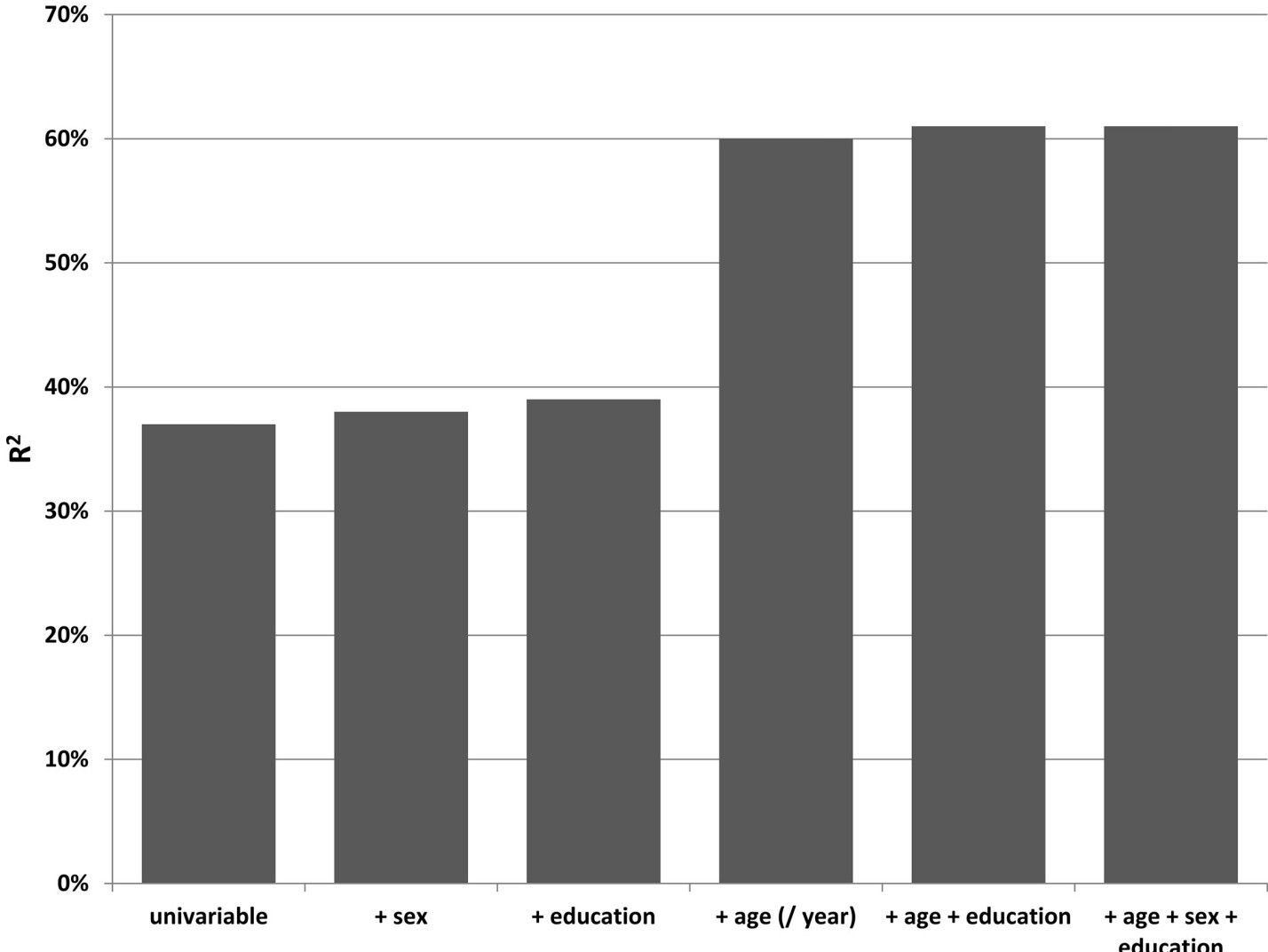

**Fig 2. Results from the linear regression analysis for the question: 'Do you have any difficulty with your hearing?'.** First univariable analysis was done, then each of the independent covariates were added, initially separately and later together. $R^2$ is given as a measure of the goodness of fit of the model.

**Table 3. Crosstab for the dichotomized answers to the question 'Do you have any difficulty with your hearing?'.** For defining mild hearing loss, all positive answers were included. For moderate hearing loss, "regularly" and "often" were the answer options included. These subjective measures were compared to the objective $PTA_{0.5-4}$, mild: $\geq 20 dBHL$, and moderate: $\geq 35 dBHL$. Number of participants are depicted in each category.

| | | Whole population | | | < 65 years | | | 65–80 years | | | > 80 years | | |
|---|---|---|---|---|---|---|---|---|---|---|---|---|---|
| **Subjective hearing loss** | | **No** | **Yes** | **Preva-lence (%)** | **No** | **Yes** | **Preva-lence (%)** | **No** | **Yes** | **Preva-lence (%)** | **No** | **Yes** | **Preva-lence (%)** |
| Mild hearing loss $\geq 20 dBHL$ | No | 1,610 | 715 | 47.4% | 911 | 468 | 77.5% | 643 | 226 | 37.6% | 56 | 21 | 9.4% |
| | Yes | 777 | 1,804 | 52.6% | 113 | 287 | 22.5% | 481 | 960 | 62.4% | 183 | 557 | 90.6% |
| Prevalence (%) | | 48.7% | 51.3% | | 57.7% | 42.3% | | 48.7% | 52.3% | | 29.3% | 70.7% | |
| Moderate hearing loss $\geq 35$ dBHL | No | 3,593 | 340 | 80.2% | 1,570 | 149 | 96.6% | 1,705 | 151 | 80.3% | 318 | 40 | 43.8% |
| | Yes | 440 | 533 | 19.8% | 17 | 43 | 3.4% | 206 | 248 | 19.7% | 217 | 242 | 56.2% |
| Prevalence (%) | | 82.2% | 17.8% | | 89.2% | 9.1% | | 82.7% | 17.3% | | 65.5% | 34.5% | |

(>80 years) has the highest AUC for mild hearing loss, which is understandable as almost all people in that age group have some form of hearing loss.

## Conclusion

Self-reported hearing loss, using the question 'Do you have any difficulty with your hearing (without hearing aids)?', has reasonable ability to detect both mild and moderate hearing loss in older adults, especially when the age of the individual is factored into the answer. This finding is mainly of importance for large population-based studies in which audiometry is absent but hearing loss still has to be quantified. A single question cannot substitute regular audiometry, but it is able to assess hearing on a population level with reasonable accuracy, adjusted for the age of the individuals.

## Supporting information

**S1 Table. Average hearing loss (in dB) for PTA$_{0.5-4}$.** For 3 age categories, sex and highest achieved level of education.
(DOCX)

## Acknowledgments

We are grateful to all the participants and the staff from the Rotterdam Study.

## Author Contributions

**Conceptualization:** Berthe C. Oosterloo, M. Arfan Ikram, André Goedegebure.

**Data curation:** Berthe C. Oosterloo, M. Arfan Ikram.

**Formal analysis:** Berthe C. Oosterloo, Nienke C. Homans.

**Funding acquisition:** Rob J. Baatenburg de Jong, André Goedegebure.

**Investigation:** Berthe C. Oosterloo, A. Paul Nagtegaal.

**Methodology:** Berthe C. Oosterloo, Nienke C. Homans, André Goedegebure.

**Supervision:** Rob J. Baatenburg de Jong, M. Arfan Ikram, A. Paul Nagtegaal, André Goedegebure.

**Writing – original draft:** Berthe C. Oosterloo, Nienke C. Homans.

**Writing – review & editing:** Rob J. Baatenburg de Jong, M. Arfan Ikram, A. Paul Nagtegaal, André Goedegebure.

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
