## [Decision Letter · Decision Letter 0]

18 Aug 2019

PONE-D-19-19122

Assessing hearing loss in older adults with a single question and simple person characteristics; comparison with pure tone audiometry in the Rotterdam Study

PLOS ONE

Dear Mrs Oosterloo,

Thank you for submitting your manuscript to PLOS ONE. After careful consideration, we feel that it has merit but does not fully meet PLOS ONE’s publication criteria as it currently stands. Therefore, we invite you to submit a revised version of the manuscript that addresses the points raised during the review process.

We would appreciate receiving your revised manuscript by Oct 02 2019 11:59PM. To enhance the reproducibility of your results, we recommend that if applicable you deposit your laboratory protocols in protocols.io, where a protocol can be assigned its own identifier (DOI) such that it can be cited independently in the future. For instructions see: http://journals.plos.org/plosone/s/submission-guidelines#loc-laboratory-protocols

We look forward to receiving your revised manuscript.

Kind regards,

Francesco Martines, PhD

Academic Editor

PLOS ONE

Journal Requirements:

1. We note that you have indicated that data from this study are available upon request. PLOS only allows data to be available upon request if there are legal or ethical restrictions on sharing data publicly. For information on unacceptable data access restrictions, please see http://journals.plos.org/plosone/s/data-availability#loc-unacceptable-data-access-restrictions.

Reviewers' comments:

Reviewer's Responses to Questions

**Comments to the Author**

1. Is the manuscript technically sound, and do the data support the conclusions?

Reviewer #1: Yes

Reviewer #2: Yes

Reviewer #3: Partly

Reviewer #4: Yes

2. Has the statistical analysis been performed appropriately and rigorously? 

Reviewer #1: Yes

Reviewer #2: Yes

Reviewer #3: I Don't Know

Reviewer #4: No

3. Have the authors made all data underlying the findings in their manuscript fully available?

Reviewer #1: Yes

Reviewer #2: Yes

Reviewer #3: Yes

Reviewer #4: Yes

4. Is the manuscript presented in an intelligible fashion and written in standard English?

Reviewer #1: No

Reviewer #2: Yes

Reviewer #3: Yes

Reviewer #4: No

5. Review Comments to the Author

Reviewer #1: Interesting paper, I would recommend being clearer as to your purpose in refining a one question hearing test as a practical way to measure hearing impairment in large populations. You make that clear in the paper but not as clear in the abstract.

Comments are based on line of paper.

24-25rewrite HL is a frequent concern in the elderly population warranting investigation in numerous cohort studies. PT audiometry, the diagnostic gold standard for hearing impairment, is time consuming and costly for large population studies.

This statement clears up confusion that you are advocating for using one question to test hearing in individuals.

29-delete s on answer

38-39 sentence is missing words

46 rewrite ...assessed with reasonable accuracy...

50 which general population is growing older? western countries?

52 dB require a reference, in this case and for the entire paper use dBHL

55 depression making it imperative to identify hearing impairment at its onset

57 Self-reporting scales regarding hearing.... statement needs references

60 reference statement about extensive investigation

63-64 using a single question format.

75 ..increasing the positive correlation

77 ..which can lead to

78 change life to listening

83 hearing loss using a single question format.

93 45 years and older

97 recent result was taken

106 be clearer about what was asked in the extensive interview. What other questions were posed could impact how person felt about general body state, mental state and state of sensory abiliites

Analysis was well explained and your stats methods were clearly stated

150 primary education levels

184 Add a sentence explaining the impact of age of individual in predicting hearing loss

185 support in the literature

187 in which the time

188 audiometry are not practical or possible (delete sparse)

190 level. One question testing, however, wil...

192 In addition, we ...

203 delete distinct replace with distinguish

204 categories of answer foils when attempting to identify mild hearing loss

207 amongst other factors,

215 older people to experience hearing impairment.

216 older individuals as compared to younger persons

218 underestimation of hearing impairment and its symptoms would be rare.

226 when age of the individual is factored into the answer

226 delete additional

227.. there is a need to quantify hearing impairment.

228 substitute for audiometry, assess

229 on a population level, hearing can be assessed reasonably accurately with a single

230 question corrected for age of individual.

Reviewer #2: Using a large data set from the Rotterdam study, the authors aim to evaluate the effectiveness of estimating the magnitude of hearing loss based on the answer to a single question. Other attempts at achieving this same general goal and their relative success and failure is discussed. The authors specifically point out three deficits or complications of previous work that the current manuscript attempts to overcome. First, previous work included significant numbers of younger individuals with normal hearing, arguably inflating correlation. Second, previous work focused on finding the relationship between self-report and moderate or more severe hearing loss. And finally, these authors attempt to examine the improvement in single question performance when individual subject characteristics such as age are included in the prediction model.

The authors demonstrate considerable success in estimating hearing loss based on the four-layer answer to the single question. Performance of the single question improves on the inclusion of age as a variable. On average hearing loss increases by ~ 10.5 dB between categories in the answer. Diagnosis of both mild and moderate hearing loss is possible based on the answer to the single question, with performance being slightly better for moderate hearing loss.

Was sex, education, and age taken from participant reports? If that was the case, then perhaps sex should be referred to gender as it is reported but not biologically verified.

Why did the authors use 45 years as the cut off age for their sample?

Other items:

Line 80, the following sentence is difficulty to interpret: “Most studies report whether certain subgroups of individuals being better or worse at predicting hearing loss.”

The discussion regarding over and underestimation of hearing loss in the second half of page 12 is a bit confusing. The authors seem to argue at first that estimates are essentially one sided in younger individuals as objective hearing loss is limited. In contrast the estimates are two sided in older individuals. But the authors also argue that older individuals should be able to better estimate their objective hearing loss as compensation is more difficult and it is more socially acceptable to have hearing loss. Given the population sample that the authors have, the above hypotheses are verifiable. It may strengthen the paper if the authors attempted some secondary analyses after subdividing the sample into older and younger groups. In the limit, the error distributions could be estimated in five- or seven-year age windows.

Reviewer #3: This paper has some issues as there is a claim that the single question provides good estimation of hearing loss (PTA) in an older population, but in fact the variance accounted for by the answers to this question is 37% and it is only when other variables, particularly age, are used as additional predictors that the statistics show a "good" prediction. It is hardly a surprise that there would be some correlation between the question and PTA in the better ear and this has been shown before as noted in the introduction to the paper. In a population sense, age is quite a good predictor of hearing loss and this is certainly well known, so I am not sure that there is anything new here. I would concede that in a large population study with limitations of funding, the one question response plus the demographic factors (particularly age) will provide a good estimate of the prevalence of mild and moderate hearing loss, but the question by itself is not a very good "predictor" of the audiogram in an individual. It is only an estimator of prevalence. I think the manuscript needs to be more carefully written so that it is clear that we do not really have a prediction here, but only an estimator of prevalence. I also found some of the statistical discussion confusing. The analysis is named as a univariable regression but seems to be describing a multiple regression. Maybe I am misunderstanding the way the authors have used the term univariable. Additional specific comments are below.

p.4, l.50 "the general population grows older" - of course, but the issue is that people are living longer

p.4, l.55 "identify hearing loss in time" - in time for what?

p.11, l.193 This is not the predictive ability of the question, it is the predictive ability of the 4 variables. What would be the predictive ability of age, gender and education without the question?

p.12, l.203 "distinct" should be 'distinguish"

p.12, l.212 "in" should be "into"

p.12, l.216 "compared as in" should be "compared to"

p.12, l.217 "might as well be attributed" should be "may be attributable"

p.13, l.228 "asses" should be "assess"

p.13, l.229 "on population level' should be "on a population level"

p.13, l.229 "pretty accurately" - maybe "reasonably accurately" would be better

Reviewer #4: Assessing hearing loss in older adults with a single question and simple person characteristics; comparison with pure tone audiometry in the Rotterdam Study

The authors posed the question: whether a self-report using a single question of hearing loss and some additional data corresponds with audiometric approach to determining hearing loss. This is not a novel question, but important to ask none-the-less.

The analysis appears to be appropriate to the address the questions, although I have a query about the dichotomising. I also am not clear what data were used? Best ear, worst ear, mean of both?

The discussion and conclusions are properly drawn from the results, although a bit more clarity of the applicability of the findings would be good.

I would have also liked to know why this question was used? What was the basis or framework? Others have used another form of words. Why would one be better than another? Why are hearing difficulties related to severity of hearing loss?

On another note, the WHO and Global Burden of Disease Group have recommended a new approach to classification. Hume, IJA, https://doi.org/10.1080/14992027.2018.1518598

It would be very valuable, seeing this papers says the value of this approach is for population/epidemiology studies, for this classification system to be applied. Not >=25 and >=40. I can see that this needs some re-analysis, but it would make this study much more valuable. Maybe this paper will not report the prevalence of HL (just focussing on the use of a single question), but a subsequent paper should ideally use the WHO/GBD classifications.

A note on the English: the manuscript is quite readable, and the intent -if not always clear- can be assumed. However, there are traces of evidence that English is not the primary language of the authors, and the structure of the Dutch language is apparent at times. In some cases may come down to differences in choices of wording, and it does not affect the clarity of the manuscript. But there is also some loose phrasing that does need to be tightened up.

I would like to see this work published, but some attention needs to be paid to the writing. If the suggestion to use another set of classifications is not accepted, then the usefulness of the study to others in the long term will be restricted.

Specific comments:

Lines 38 and 39: spare space before the comma in line 38, and ‘are’ should be ‘area’

Line 45: asses should be assess; and ‘population’ should be ‘a population’.

Line 44: I’m not sure this statement (“never”) can be made: sure, this single question will not be a substitute. They are assess two different things. But it is possible that someone will develop a single question that will be better; but we do not know. In any case, I would leave this matter for the discussion, not a conclusion in the abstract.

Line 46: “pretty accurately” is not appropriate wording

Line 50: “The general population grows older” - taken literally, this can’t be otherwise. However, isn’t the point that the mean age of the population is increasing – and even then that is not in every country or population.

Line 57: references?

Line 67: What is ‘this’ review? Is it (14)? If so, the sentence needs some attention. This reference is a bit dated, and there have been more studies since then.

Line 75: Why is it easier? Is it easier to correct? Then this needs to be stated.

Line 82: ‘simple’ – this is a bit of a vague term. Why is one characteristic simple, and another characteristic not simple?

Line 84: ‘aim was’

Line 85: why these two levels? See earlier comment.

Line 97: “the most recent” meaning all the data (question and audiogram)? Adding the word ‘data’ after ‘recent’ (and changing ‘was’ to ‘were’) may be helpful.

Line 104: What additional data were obtained from treating physicians, and why. Treatment is not part of the study is it?

Line 107: perhaps it need not be stated, but I assume the questions and answers were in Dutch and not English.

Line 107: What is the basis for these answers? And why this particular question? This is not addressed in the introduction. I realise that this may be difficult to answer, and I doubt other similar studies have

provided a rationale either. (See comment made at the start.)

Line 117: maximum of the audiometer?

Line 118: why +5dB? And does that mean if no response was obtained at 50dB and testing continued to the maximum of the audiometer at that frequency (say 90dB) then the ‘threshold’ was set to 95dB? This is a tricky thing to manage. It is not a threshold, and who is to say they would not have heard a stimulus at 110dB? However, it’s good that this information is provided.

Line 120: ‘we used a mid-frequency average’ – ‘average’ implies one or the two ears; which ear? If both were in the analysis, then it should be ‘we used mid-frequency averages’. I did not find this clarified in the manuscript. Was the better ear data used? This needs to be stated, also on the figures.

Line 128: ‘independent variables’ is better.

Line 130: This raises the question: why did you then bother with four answers? Is it valid to combine the two pairs of answers?

Line 143: Is this needed here? I think I saw this mentioned earlier.

Line 150: “with primary education” this is hard understand without looking at the table (and even then it is not clear). Do you mean “with education only to a primary school level”?

Line 159: be clear that education and sex are added to the basic model, not in addition to age (second last row in Suppl table 2). By the way, this table is quite important for the main manuscript.

‘improved’ do you mean from 0.6 to 0.61? Because then you say it was almost unchanged.

Line 163: these figures are not shown in the table.

Line 170: I am not sure what is meant by ‘a clinical situation’

Line 179: the comma is not needed.

Line 180: ‘hearing loss’ should be ‘the severity of hearing loss’?

Line 182: What does ‘sufficient’ mean? This is undefined.

Line 190: this sentence needs some work. What is a ‘personal level’? I think what is meant that a single question may be a good proxy for hearing loss in population studies, but for clinical use it cannot replace audiometry. (However, doesn’t it have a place in clinical setting? See Swanepoel, JAAA, 2013)

Line 192: assess, not asses (also line 228)

Line 202: But weren’t the answers dichotomised when calculating sensitivity and specificity etc.?

Line 207: Do you mean on a personal level – or on a population level? I guess that the following references suggest this is dependent on the nature of the population?

Line 213: ‘recognise’ is different to being ‘socially acceptable’, isn’t it? Surely younger people are equally able to recognise that they have a hearing loss? And don’t you mean that it is more socially acceptable for older to admit that they have a hearing loss. ( I suggest dropping the word ‘suffer’ – the deaf community say that they do not ‘suffer’)

Line 218: Still have a bit of a problem with this. See previous comment. True, there is less chance that this will occur. This line with the next line needs to be tightened up a bit.

Line 219: I assume that males are not as reliable; but these data are not provided, and this sentence does not clarify it at all.

Figure 2: what does b. provide that a. does not, other than a different vertical axis? 2a: the purpose of the different symbols is not clear. I see not logic in the choice of shape or ‘colour’ or border.

Line 228: see earlier comment

Line 229: ‘pretty’ is not a word that should be used. See earlier comment.

6. PLOS authors have the option to publish the peer review history of their article (what does this mean?). If published, this will include your full peer review and any attached files.

Reviewer #1: Yes: Holly S. Kaplan

Reviewer #2: No

Reviewer #3: No

Reviewer #4: No

---

## [Author Response · Author response to Decision Letter 0]

7 Nov 2019

The complete rebuttal letter was uploaded as a document with the manuscript.

Date: 27-09-2019

Subject: Revision of manuscript PONE-D-19-19122

Dear Dr. Fransesco Martines,

Thank you for reviewing our manuscript entitled ‘Assessing hearing loss in older adults with a single question and simple person characteristics; comparison with pure tone audiometry in the Rotterdam Study’ (PONE-D-19-19122).

We are grateful for all the detailed comments by the reviewers. In response to those comments, we have made changes to the manuscript. Regarding the language issues raised by the reviewers, we asked a native speaker to adjust the language. We feel that these changes have improved the manuscript and its readability. 

Please find attached our point-by-point reply to the questions and comments of the reviewers. We hope you will consider this revised manuscript acceptable for publication in Plos One.

Yours sincerely,

On behalf of all authors,

Neelke Oosterloo, MD

Paul Nagtegaal, MD, PhD

André Goedegebure, Ir. PhD

 

Reviewer #1: 

Interesting paper, I would recommend being clearer as to your purpose in refining a one question hearing test as a practical way to measure hearing impairment in large populations. You make that clear in the paper but not as clear in the abstract.

Response: We thank the reviewer for the comments. We have modified the abstract and hope this message is clearer now. We changed the abstract according the specific suggestions you made below.

Comments are based on line of paper.

24-25rewrite HL is a frequent concern in the elderly population warranting investigation in numerous cohort studies. PT audiometry, the diagnostic gold standard for hearing impairment, is time consuming and costly for large population studies.

This statement clears up confusion that you are advocating for using one question to test hearing in individuals.

Response: We have changed the statement as suggested by the reviewer. 

38-39 sentence is missing words

Response: We hope the sentence is better this way.

50 which general population is growing older? western countries?

Response: We agree with the reviewer that this statement is unclear, therefor we changed it to:

‘Generally, world-wide life expectancy increases, resulting in an increase of age-related health problems, including hearing loss.’ 

52 dB require a reference, in this case and for the entire paper use dB HL

Response: We agree with the reviewer to add a reference to the dB. We have changed it throughout the manuscript when applicable. 

55 depression making it imperative to identify hearing impairment at its onset

Response: We agree with the reviewer that the emphasis of this sentence is not correct. We changed the sentence. 

57 Self-reporting scales regarding hearing.... statement needs references

Response: We changed the sentence and added the references.

60 reference statement about extensive investigation

Response: The requested references are listed in the sentences to follow. 

106 be clearer about what was asked in the extensive interview. What other questions were posed could impact how person felt about general body state, mental state and state of sensory abilities

Response: We added extra information to the main text. However due to the extensiveness of the interview it is too much to explain it all in this paper, so a reference was added (1).

Analysis was well explained and your stats methods were clearly stated

184 Add a sentence explaining the impact of age of individual in predicting hearing loss

Response: We have altered the paragraph in the discussion section on this to add this explanation. 

204 categories of answer foils when attempting to identify mild hearing loss

Response: We do not understand this comment. Is it possible to clarify?

Textual changes: 

29-delete -s on answer

46 rewrite ...assessed with reasonable accuracy...

63-64 using a single question format.

75 ..increasing the positive correlation

77 ..which can lead to

78 change life to listening

83 hearing loss using a single question format.

93 45 years and older

97 recent result was taken

150 primary education levels

185 support in the literature 

187 in which the time

188 audiometry are not practical or possible (delete sparse)

190 level. One question testing, however, wil...

192 In addition, we ...

203 delete distinct replace with distinguish

207 amongst other factors,

215 older people to experience hearing impairment.

216 older individuals as compared to younger persons

218 underestimation of hearing impairment and its symptoms would be rare.

226 when age of the individual is factored into the answer

226 delete additional

227.. there is a need to quantify hearing impairment.

228 substitute for audiometry, assess

229 on a population level, hearing can be assessed reasonably accurately with a single

230 question corrected for age of individual.

Response: We thank the reviewer for the suggestions to improve the quality of the language. We altered the text as suggested by the reviewer.

 

Reviewer #2:

Using a large data set from the Rotterdam study, the authors aim to evaluate the effectiveness of estimating the magnitude of hearing loss based on the answer to a single question. Other attempts at achieving this same general goal and their relative success and failure is discussed. The authors specifically point out three deficits or complications of previous work that the current manuscript attempts to overcome. First, previous work included significant numbers of younger individuals with normal hearing, arguably inflating correlation. Second, previous work focused on finding the relationship between self-report and moderate or more severe hearing loss. And finally, these authors attempt to examine the improvement in single question performance when individual subject characteristics such as age are included in the prediction model.

The authors demonstrate considerable success in estimating hearing loss based on the four-layer answer to the single question. Performance of the single question improves on the inclusion of age as a variable. On average hearing loss increases by ~ 10.5 dB between categories in the answer. Diagnosis of both mild and moderate hearing loss is possible based on the answer to the single question, with performance being slightly better for moderate hearing loss.

Response: We thank the reviewer for the comments. Please see our response to each comment below.

Was sex, education, and age taken from participant reports? If that was the case, then perhaps sex should be referred to gender as it is reported but not biologically verified.

Response: Sex was not taken from the participant report, it was objectified at the research center. We therefore choose to keep the term sex over gender. 

Why did the authors use 45 years as the cut off age for their sample?

Response: This cut off point for age is due to the study design. The Rotterdam Study is designed to investigate determinants of health and disease in an ageing population(1). All participants of the Rotterdam Study were eligible to be included in the present study even though the youngest participant in the present study was 51.4 years old (table 1).

Other items:

Line 80, the following sentence is difficulty to interpret: “Most studies report whether certain subgroups of individuals being better or worse at predicting hearing loss.”

Response: We understand the difficulty, we adjusted the sentence hoping that it is more easily interpretable.

Introduction: ‘Most studies report whether certain subgroups of individuals are better or worse at judging their hearing capacity’.

The discussion regarding over and underestimation of hearing loss in the second half of page 12 is a bit confusing. The authors seem to argue at first that estimates are essentially one sided in younger individuals as objective hearing loss is limited. In contrast the estimates are two sided in older individuals. But the authors also argue that older individuals should be able to better estimate their objective hearing loss as compensation is more difficult and it is more socially acceptable to have hearing loss. Given the population sample that the authors have, the above hypotheses are verifiable. It may strengthen the paper if the authors attempted some secondary analyses after subdividing the sample into older and younger groups. In the limit, the error distributions could be estimated in five- or seven-year age windows.

Response: We agree with the reviewer that the proposed secondary analyses will strengthen the paper, we added them to the results section. However, we were not able to assess this in five or seven year age windows, as certain groups would contain a relatively low number of participants.(2)

 

Reviewer #3: 

This paper has some issues as there is a claim that the single question provides good estimation of hearing loss (PTA) in an older population, but in fact the variance accounted for by the answers to this question is 37% and it is only when other variables, particularly age, are used as additional predictors that the statistics show a "good" prediction. It is hardly a surprise that there would be some correlation between the question and PTA in the better ear and this has been shown before as noted in the introduction to the paper. In a population sense, age is quite a good predictor of hearing loss and this is certainly well known, so I am not sure that there is anything new here. I would concede that in a large population study with limitations of funding, the one question response plus the demographic factors (particularly age) will provide a good estimate of the prevalence of mild and moderate hearing loss, but the question by itself is not a very good "predictor" of the audiogram in an individual. It is only an estimator of prevalence. I think the manuscript needs to be more carefully written so that it is clear that we do not really have a prediction here, but only an estimator of prevalence. I also found some of the statistical discussion confusing. 

Response: We thank the reviewer for the comments on the manuscript. We agree that it is not new to show that age is a good predictor for hearing loss in a population sense, just like sex and hearing loss. Neither is it new that a single question format to assess hearing loss is able to estimate the prevalence. However, this study is new in a sense that we combine these 4 factors to assess hearing loss on a population basis. We agree that one of the possible applications of such a question is to estimate prevalence of hearing loss in a large population. However, the question could also be used to identify subgroups of a large population that could be compared, e.g. to analyze genetic susceptibility for hearing loss (GWAS).

The analysis is named as a univariable regression but seems to be describing a multiple regression. Maybe I am misunderstanding the way the authors have used the term univariable. 

Response: The analyses started with a true univariable analysis (dependent variable: PTA0.5,1,2,4,, independent variable: the interview answers) , after that the covariates were added and we do not speak of a univariable analysis anymore. 

Additional specific comments are below.

p.4, l.50 "the general population grows older" - of course, but the issue is that people are living longer

Response: This statement was changed into: ‘In the population of western countries the life expectancy increases’ we feel that with this change in sentence the issue of longevity and an increasing prevalence is captured.

p.4, l.55 "identify hearing loss in time" - in time for what?

Response: We see that the statement is unclear, the sentences were changed into:” Its impact is substantial, as it is associated with social withdrawal, cognitive decline and depression.”

p.11, l.193 This is not the predictive ability of the question, it is the predictive ability of the 4 variables. What would be the predictive ability of age, gender and education without the question?

Response: We know that these factors by itself have a high predictive ability for hearing loss, without the question the AUC is 0.82, which shows us that the question does contribute to the estimation of hearing loss (AUC:0.86).

Textual changes: 

p.12, l.203 "distinct" should be 'distinguish"

p.12, l.212 "in" should be "into"

p.12, l.216 "compared as in" should be "compared to"

p.12, l.217 "might as well be attributed" should be "may be attributable"

p.13, l.228 "asses" should be "assess"

p.13, l.229 "on population level' should be "on a population level"

p.13, l.229 "pretty accurately" - maybe "reasonably accurately" would be better

Response: We thank the reviewer for the suggestions to improve the quality of the language. We altered the text as suggested by the reviewer

 

Reviewer #4: 

Assessing hearing loss in older adults with a single question and simple person characteristics; comparison with pure tone audiometry in the Rotterdam Study

The authors posed the question: whether a self-report using a single question of hearing loss and some additional data corresponds with audiometric approach to determining hearing loss. This is not a novel question, but important to ask none-the-less.

The analysis appears to be appropriate to the address the questions, although I have a query about the dichotomising. I also am not clear what data were used? Best ear, worst ear, mean of both?

Response: We thank the reviewer for the comments. We can see that there was too little information, we used the best ear, this we also added to the method section to clarify. 

The discussion and conclusions are properly drawn from the results, although a bit more clarity of the applicability of the findings would be good.

Response: We have added 

I would have also liked to know why this question was used? What was the basis or framework? Others have used another form of words. Why would one be better than another? Why are hearing difficulties related to severity of hearing loss?

Response: In the time that the data collection on hearing was designed, around 2010/2011, a review of the literature was available on subjective hearing assessment by Chou et al. (2). This review showed a nice overview of several studies that used questions varying between ‘Would you say that you have any difficulty hearing? / Do you have difficulty hearing / Do you feel you have hearing loss?/ Do you have a hearing problem now?’

On another note, the WHO and Global Burden of Disease Group have recommended a new approach to classification. Hume, IJA, https://doi.org/10.1080/14992027.2018.1518598

It would be very valuable, seeing this papers says the value of this approach is for population/epidemiology studies, for this classification system to be applied. Not >=25 and >=40. I can see that this needs some re-analysis, but it would make this study much more valuable. Maybe this paper will not report the prevalence of HL (just focussing on the use of a single question), but a subsequent paper should ideally use the WHO/GBD classifications.

Response: We thank the reviewer for bringing up this discussion on the cut-off points. We agree with the reviewer that these cut-off points are better supported by literature. All analyses were rerun with these adjusted cut-off points, resulting in marginally different results. 

A note on the English: the manuscript is quite readable, and the intent -if not always clear- can be assumed. However, there are traces of evidence that English is not the primary language of the authors, and the structure of the Dutch language is apparent at times. In some cases may come down to differences in choices of wording, and it does not affect the clarity of the manuscript. But there is also some loose phrasing that does need to be tightened up.

Response: We thank the reviewer for his/her comments and adjustments on the writing. The manuscript was redirected by a native speaking person. 

I would like to see this work published, but some attention needs to be paid to the writing. If the suggestion to use another set of classifications is not accepted, then the usefulness of the study to others in the long term will be restricted.

Specific comments:

Textual changes:

Lines 38 and 39: spare space before the comma in line 38, and ‘are’ should be ‘area’

Line 45: asses should be assess; and ‘population’ should be ‘a population’.

Line 46: “pretty accurately” is not appropriate wording

Line 84: ‘aim was’

Line 128: ‘independent variables’ is better.

Line 150: “with primary education” this is hard understand without looking at the table (and even then it is not clear). Do you mean “with education only to a primary school level”?

Line 179: the comma is not needed.

Line 180: ‘hearing loss’ should be ‘the severity of hearing loss’?

Line 192: assess, not asses (also line 228)

Line 228: see earlier comment

Line 229: ‘pretty’ is not a word that should be used. See earlier comment.

Response: We thank the reviewer for the suggestions to improve the quality of the language. We altered the text as suggested by the reviewer.

Line 44: I’m not sure this statement (“never”) can be made: sure, this single question will not be a substitute. They are assess two different things. But it is possible that someone will develop a single question that will be better; but we do not know. In any case, I would leave this matter for the discussion, not a conclusion in the abstract.

Response: We agree that this a too strict statement, we changed ‘will never’ to ‘cannot’.

Line 50: “The general population grows older” - taken literally, this can’t be otherwise. However, isn’t the point that the mean age of the population is increasing – and even then that is not in every country or population.

Response: We agree that this statement is not correct, we changed it for ‘In the population of western countries the life expectancy increases’. 

Line 57: references?

Response: We added the references to this sentence.

Line 67: What is ‘this’ review? Is it (14)? If so, the sentence needs some attention. This reference is a bit dated, and there have been more studies since then.

Response: We deleted the sentence between the reference and ‘this review’. A newer reference was added.

Line 75: Why is it easier? Is it easier to correct? Then this needs to be stated.

Response: It is generally easier because the hearing of younger participants is good in general, so if most do not have any objective hearing problems it is to be expected that they consider it subjectively good as well. We added ‘as this is generally good and will be adequately reported as no hearing problems.’ to the sentence in the main text, we hope this clarifies the statements. 

Line 82: ‘simple’ – this is a bit of a vague term. Why is one characteristic simple, and another characteristic not simple?

Response: We agree that simple is a vague term, therefore we changed it to easily obtainable as this is what was meant with simple.

Line 85: why these two levels? See earlier comment.

Response: We agree with the reviewer and have changed this throughout the manuscript.

Line 97: “the most recent” meaning all the data (question and audiogram)? Adding the word ‘data’ after ‘recent’ (and changing ‘was’ to ‘were’) may be helpful.

Response: We agree with the reviewer and have added ‘results’ to clarify. 

Line 104: What additional data were obtained from treating physicians, and why. Treatment is not part of the study is it?

Response: We removed this sentence from the manuscript as these data were not used in this current study.

Line 107: perhaps it need not be stated, but I assume the questions and answers were in Dutch and not English.

Response: The questions were in Dutch. 

Line 107: What is the basis for these answers? And why this particular question? This is not addressed in the introduction. I realise that this may be difficult to answer, and I doubt other similar studies have

provided a rationale either. (See comment made at the start.)

Response: See explanation above.

Line 117: maximum of the audiometer?

Response: This depends on the frequency. We have changed this in the methods section to maximum stimulation level of the audiometer for that given frequency.

Line 118: why +5dB? And does that mean if no response was obtained at 50dB and testing continued to the maximum of the audiometer at that frequency (say 90dB) then the ‘threshold’ was set to 95dB? This is a tricky thing to manage. It is not a threshold, and who is to say they would not have heard a stimulus at 110dB? However, it’s good that this information is provided.

Response: This is a standard procedure in audiometry used in research, as a threshold is warranted to be able to execute the calculations. At a given frequency if the maximum stimulus in not heard the threshold is set 5dB above the maximum stimulus. 

Line 120: ‘we used a mid-frequency average’ – ‘average’ implies one or the two ears; which ear? If both were in the analysis, then it should be ‘we used mid-frequency averages’. I did not find this clarified in the manuscript. Was the better ear data used? This needs to be stated, also on the figures.

Response: We added this to the manuscript and figures. The better ear data was used.

Line 130: This raises the question: why did you then bother with four answers? Is it valid to combine the two pairs of answers?

Response: Four answer categories were used to determine some degree of hearing loss. In the manuscript we show that considering these four answer categories one can differentiate between mild and moderate hearing loss by using different cut-off points. We think it is valid to combine the pairs of answers, this can also be derived from figure 1. 

Line 143: Is this needed here? I think I saw this mentioned earlier.

Response: It should be added somewhere in the manuscript. We added a subheading to this section.

Line 159: be clear that education and sex are added to the basic model, not in addition to age (second last row in Suppl table 2). By the way, this table is quite important for the main manuscript.

‘improved’ do you mean from 0.6 to 0.61? Because then you say it was almost unchanged.

Response: We have added this table to the main manuscript and removed figure 2a, as these show the same results in a different way. We tested the fit of the regression model with a likelihood ratio test, this showed a significant improvement of the model although the explained variance (R2) remained almost unchanged, as stated in the manuscript.

Line 163: these figures are not shown in the table.

Response: We have shown the raw numbers in the table. We added the prevalence data to the table, sensitivity, specificity, negative predictive value and positive predictive value are described in the text. 

Line 170: I am not sure what is meant by ‘a clinical situation’

Response: We agree with the reviewer that is unclear, we removed this part of the sentence. 

Line 182: What does ‘sufficient’ mean? This is undefined.

Response: Sufficient was changed to reasonable.

Line 190: this sentence needs some work. What is a ‘personal level’? I think what is meant that a single question may be a good proxy for hearing loss in population studies, but for clinical use it cannot replace audiometry. (However, doesn’t it have a place in clinical setting? See Swanepoel, JAAA, 2013)

Response: We agree that we think that a single question can be used as a proxy in large population based studies, although it cannot replace pure tone audiometry in an individual. We do think it might have a place in a clinical setting for screening purposes, but it is (for now) not as accurate as pure tone audiometry, and therefore cannot replace pure tone audiometry in a diagnostic setting. 

Line 202: But weren’t the answers dichotomised when calculating sensitivity and specificity etc.?

Response: Yes, the answers were dichotomized, however, we think that the dichotomization of the four answer categories contribute to the ability to detect mild or moderate hearing loss.

Line 207: Do you mean on a personal level – or on a population level? I guess that the following references suggest this is dependent on the nature of the population?

Response: This is correct, both on a personal and a population level.

Line 213: ‘recognise’ is different to being ‘socially acceptable’, isn’t it? Surely younger people are equally able to recognise that they have a hearing loss? And don’t you mean that it is more socially acceptable for older to admit that they have a hearing loss. ( I suggest dropping the word ‘suffer’ – the deaf community say that they do not ‘suffer’)

Response: We agree that recognize is not the best chosen word. We changed it for ‘report’ as we think that this covers the message better. We removed suffer from the text, thank you for pointing this out.

Line 218: Still have a bit of a problem with this. See previous comment. True, there is less chance that this will occur. This line with the next line needs to be tightened up a bit.

Response: We adjusted the two lines in: ‘As hearing is generally good in younger people, underestimation of hearing impairment and its symptoms would be rare. Whereas in older people with more prevalent hearing loss, both under- and over estimation are possible.’

Line 219: I assume that males are not as reliable; but these data are not provided, and this sentence does not clarify it at all.

Response: We realized that these data were not provided in the results section, we therefore dropped the sentences. 

Figure 2: what does b. provide that a. does not, other than a different vertical axis? 2a: the purpose of the different symbols is not clear. I see not logic in the choice of shape or ‘colour’ or border.

Response: We removed figure 2a from the main text and substituted it with table 2. 

References:

1. Ikram MA, Brusselle GGO, Murad SD, van Duijn CM, Franco OH, Goedegebure A, et al. The Rotterdam Study: 2018 update on objectives, design and main results. Eur J Epidemiol. 2017;32(9):807-50.

2. Chou R, Dana T, Bougatsos C, Fleming C, Beil T. Screening adults aged 50 years or older for hearing loss: a review of the evidence for the U.S. preventive services task force. Ann Intern Med. 2011;154(5):347-55.

---

## [Decision Letter · Decision Letter 1]

25 Nov 2019

PONE-D-19-19122R1

Assessing hearing loss in older adults with a single question and simple person characteristics; comparison with pure tone audiometry in the Rotterdam Study

PLOS ONE

Dear Mrs Oosterloo,

Thank you for submitting your manuscript to PLOS ONE. After careful consideration, we feel that it has merit but does not fully meet PLOS ONE’s publication criteria as it currently stands. Therefore, we invite you to submit a revised version of the manuscript that addresses the points raised during the review process.

We would appreciate receiving your revised manuscript by Jan 09 2020 11:59PM. To enhance the reproducibility of your results, we recommend that if applicable you deposit your laboratory protocols in protocols.io, where a protocol can be assigned its own identifier (DOI) such that it can be cited independently in the future. For instructions see: http://journals.plos.org/plosone/s/submission-guidelines#loc-laboratory-protocols

We look forward to receiving your revised manuscript.

Kind regards,

Francesco Martines, PhD

Academic Editor

PLOS ONE

Reviewers' comments:

Reviewer's Responses to Questions

**Comments to the Author**

1. If the authors have adequately addressed your comments raised in a previous round of review and you feel that this manuscript is now acceptable for publication, you may indicate that here to bypass the “Comments to the Author” section, enter your conflict of interest statement in the “Confidential to Editor” section, and submit your "Accept" recommendation.

Reviewer #1: All comments have been addressed

Reviewer #2: All comments have been addressed

Reviewer #3: (No Response)

Reviewer #4: All comments have been addressed

2. Is the manuscript technically sound, and do the data support the conclusions?

Reviewer #1: Yes

Reviewer #2: Yes

Reviewer #3: Partly

Reviewer #4: Yes

3. Has the statistical analysis been performed appropriately and rigorously? 

Reviewer #1: Yes

Reviewer #2: Yes

Reviewer #3: Yes

Reviewer #4: Yes

4. Have the authors made all data underlying the findings in their manuscript fully available?

Reviewer #1: Yes

Reviewer #2: No

Reviewer #3: Yes

Reviewer #4: No

5. Is the manuscript presented in an intelligible fashion and written in standard English?

Reviewer #1: Yes

Reviewer #2: Yes

Reviewer #3: Yes

Reviewer #4: Yes

6. Review Comments to the Author

Reviewer #1: Authors have addressed all of my concerns. I believe the document should be published. I congratulate them on an interesting paper.

Reviewer #2: Thanks for addressing the reviewer concerns. The paper may still benefit from some editorial overview.

Reviewer #3: The authors have responded reasonably to my comments. I still feel that there is a tendency to overclaim the effectiveness of the "single question' in identifying hearing loss. In the author's response to one of my comments, it is noted that age, gender and education have high predictability for hearing loss (without the question) with an AUC (area under the ROC curve) of 0.82. This does improve to 0.86 with the use of the question but it suggests that the prediction is not greatly improved. i feel this information should be included in the paper.

Reviewer #4: A clean copy (not showing changes) has not been provided; the initially submitted version was instead. Furthermore, there were two versions of the figures.

The authors appear to have tackled all the comments and suggestions that I and the other reviewers made. However, I suggest another careful review of the English a few errors in wording and sentence structure are still there.

Line 77: capital S

Line 79: Not sure this captures the matter correctly. And the sentence swaps from the prevalence of hearing loss in younger adults (population) to their own hearing (an individual). I agree that ease of estimation is judged on a personal level. But the wording suggests that it is easier to estimate good hearing than poor(er) hearing. Is that the case? And that the ease translates to better estimation of prevalence. This section needs some work. I can see what the authors are getting at, but I am not convinced. And is this not complicated with the issue noted in the Discussion in Lines 267 onwards?

Line 84: Starting with ‘While’ leads the reader to expect more in the sentence. I suggest dropping the word.

Line 89: ‘have’ is correct, not ‘has’

Line 121: I know that for the previous review it was not necessary to point out that the question and answers were in Dutch. However, upon reflection it would be good to make that clear, in case someone uses the same wording, and then compares results; this would not be entirely appropriate as the English translation/version has not been validated.

Line 133: is it really not possible to reference this?

Table 1: The age range also needs to have ‘years’ in the first column

Line 177: ‘Hearing thresholds were’, not ‘Hearing loss was’

Line 181: ‘thresholds’? But actually it’s PTA.5-4 isn’t it?

Lines 188 to 195; two versions of the table caption.

Table 3: The placement of the words ‘Prevalence’ is a bit unhelpful when the 100% figures are seen. Can these 100% figures be removed?

Line 240: This needs to be more nuanced. To start with, I do not think anyone would think it would replace clinical assessment of individuals, which this implies. On the other hand, I think it may have some clinical value in identifying people who are at a higher risk of hearing loss, and so this information has potential for use as a screening tool.

Line 259 onwards: Comment/question: were males better or worse at estimation than females? This matter of self-reporting is complicated by things like denial of hearing problems and self-awareness which may be lower in males, but on the other hand males having a higher prevalence of HL. I think the authors have grappled with some of these issues.

Line 266: points?

7. PLOS authors have the option to publish the peer review history of their article (what does this mean?). If published, this will include your full peer review and any attached files.

Reviewer #1: No

Reviewer #2: No

Reviewer #3: Yes: Richard C Dowell

Reviewer #4: Yes: Robert Henry Eikelboom

---

## [Author Response · Author response to Decision Letter 1]

30 Dec 2019

Reviewer #1: Authors have addressed all of my concerns. I believe the document should be published. I congratulate them on an interesting paper.

Response: We thank the reviewer for his/her work.

Reviewer #2: Thanks for addressing the reviewer concerns. The paper may still benefit from some editorial overview.

Response: We thank the reviewer for his/her work and input, we have carefully gone through the text again.

Reviewer #3: The authors have responded reasonably to my comments. I still feel that there is a tendency to overclaim the effectiveness of the "single question' in identifying hearing loss. In the author's response to one of my comments, it is noted that age, gender and education have high predictability for hearing loss (without the question) with an AUC (area under the ROC curve) of 0.82. This does improve to 0.86 with the use of the question but it suggests that the prediction is not greatly improved. i feel this information should be included in the paper.

Response: We thank the reviewer for his work. We agree this is not an huge increase, however the scope of our paper is to investigate the question as the main outcome. The additional information in this paper is that the ability to identify hearing loss can be improved when age is taken into account on a population level. 

Reviewer #4: A clean copy (not showing changes) has not been provided; the initially submitted version was instead. Furthermore, there were two versions of the figures.

The authors appear to have tackled all the comments and suggestions that I and the other reviewers made. However, I suggest another careful review of the English a few errors in wording and sentence structure are still there.

Response: We thank the reviewer for his careful review of the manuscript. We hope that we have sufficiently addressed the issues raised by the reviewer. 

Line 77: capital S

Response: We have changed this.

Line 79: Not sure this captures the matter correctly. And the sentence swaps from the prevalence of hearing loss in younger adults (population) to their own hearing (an individual). I agree that ease of estimation is judged on a personal level. But the wording suggests that it is easier to estimate good hearing than poor(er) hearing. Is that the case? And that the ease translates to better estimation of prevalence. This section needs some work. I can see what the authors are getting at, but I am not convinced. And is this not complicated with the issue noted in the Discussion in Lines 267 onwards?

Response: We thank the reviewer for this point. What we aim to point out here is that statistics of self-reported hearing loss in a younger group is highly influenced by the relatively low prevalence of hearing loss, and should not be compared with the statistics of an older population. As this point has been addressed in more detail in the discussion, we shortened and rephrased the text in this section. 

Line 84: Starting with ‘While’ leads the reader to expect more in the sentence. I suggest dropping the word.

Response: We have dropped the word while.

Line 89: ‘have’ is correct, not ‘has’

Response: We have changed has to have.

Line 121: I know that for the previous review it was not necessary to point out that the question and answers were in Dutch. However, upon reflection it would be good to make that clear, in case someone uses the same wording, and then compares results; this would not be entirely appropriate as the English translation/version has not been validated.

Response: We have added to the main text that the question was in Dutch.

Line 133: is it really not possible to reference this?

Response: Unfortunately not. This is done in all studies to be able to calculate with the hearing thresholds when the threshold of a participant is above the maximum stimulation level.

Table 1: The age range also needs to have ‘years’ in the first column

Response: We have added this to the table.

Line 177: ‘Hearing thresholds were’, not ‘Hearing loss was’

Response: We have changed this.

Line 181: ‘thresholds’? But actually it’s PTA.5-4 isn’t it?

Response: We have changed the sentence to ‘…increase of the PTA0.5-4 hearing threshold…’ as we feel this reflects the result best and meets the suggestion of the reviewer.

Lines 188 to 195; two versions of the table caption.

Response: We see what the reviewer means, but the second capitation is for figure 2.

Table 3: The placement of the words ‘Prevalence’ is a bit unhelpful when the 100% figures are seen. Can these 100% figures be removed?

Response: We have removed the 100% figure.

Line 240: This needs to be more nuanced. To start with, I do not think anyone would think it would replace clinical assessment of individuals, which this implies. On the other hand, I think it may have some clinical value in identifying people who are at a higher risk of hearing loss, and so this information has potential for use as a screening tool.

Response: We agree with the reviewer there are different perspectives on how the results can be interpreted. Therefore, we have chosen to emphasize the value of this tool in population based studies. The statement about the clinical value has been rephrased in a way that the perspective of screening has been taken in account, according to the reviewers comment.

Line 259 onwards: Comment/question: were males better or worse at estimation than females? This matter of self-reporting is complicated by things like denial of hearing problems and self-awareness which may be lower in males, but on the other hand males having a higher prevalence of HL. I think the authors have grappled with some of these issues.

Response: Males were better at the estimation than females, in spite of many other possible factors we think that the higher prevalence of hearing loss in males might have played an important role explaining this difference.

Line 266: points?

 Response: We have changed point to points.

---

## [Decision Letter · Decision Letter 2]

14 Jan 2020

Assessing hearing loss in older adults with a single question and person characteristics; comparison with pure tone audiometry in the Rotterdam Study

PONE-D-19-19122R2

Dear Dr. Oosterloo,

We are pleased to inform you that your manuscript has been judged scientifically suitable for publication and will be formally accepted for publication once it complies with all outstanding technical requirements.

With kind regards,

Francesco Martines, PhD

Academic Editor

PLOS ONE

Additional Editor Comments (optional):

Reviewers' comments:

Reviewer's Responses to Questions

**Comments to the Author**

1. If the authors have adequately addressed your comments raised in a previous round of review and you feel that this manuscript is now acceptable for publication, you may indicate that here to bypass the “Comments to the Author” section, enter your conflict of interest statement in the “Confidential to Editor” section, and submit your "Accept" recommendation.

Reviewer #3: All comments have been addressed

2. Is the manuscript technically sound, and do the data support the conclusions?

Reviewer #3: Yes

3. Has the statistical analysis been performed appropriately and rigorously? 

Reviewer #3: Yes

4. Have the authors made all data underlying the findings in their manuscript fully available?

Reviewer #3: Yes

5. Is the manuscript presented in an intelligible fashion and written in standard English?

Reviewer #3: Yes

6. Review Comments to the Author

Reviewer #3: (No Response)

7. PLOS authors have the option to publish the peer review history of their article (what does this mean?). If published, this will include your full peer review and any attached files.

Reviewer #3: Yes: Richard C Dowell

---

## [Editor Report · Acceptance letter]

17 Jan 2020

PONE-D-19-19122R2 

Assessing hearing loss in older adults with a single question and person characteristics; comparison with pure tone audiometry in the Rotterdam Study 

Dear Dr. Oosterloo:

I am pleased to inform you that your manuscript has been deemed suitable for publication in PLOS ONE. Congratulations! Your manuscript is now with our production department. 

With kind regards,

on behalf of

Dr. Francesco Martines 

Academic Editor

PLOS ONE